# Whistleblowing in Norwegian Municipalities—Can Offers of Reward Influence Employees' Willingness and Motivation to Report Wrongdoings?

**Jarle Løwe Sørensen \*, Ann Mari Nilsen Gaup and Leif Inge Magnussen**

USN School of Business, University of South-Eastern Norway, 3603 Kongsberg, Norway;
   annmarigaup@gmail.com (A.M.N.G.); Leif.Magnussen@usn.no (L.I.M.)

**\*** Correspondence: jarle.sorensen@usn.no

**Abstract:** This organizational study aims to explore whistleblowing in Norwegian Municipalities. The purpose is to explore whether employees perceive that their workplace has a well-functioning reporting system, to investigate what kind of rewards, if any, the employees considered most desirable, and to map, if any, the relationship between all types of compensation and the willingness to notify within one's own organization. This study reports on 2018 interview data from a medium-sized Norwegian municipality. The main findings indicated that the municipally exhibits little perceived every-day focus on fighting corruption and that the employees have limited knowledge of the systems and routines available to them. Further, results showed that multiple factors influenced the employee's willingness to report and receive compensation. Especially was increased management recognition and a more clearly formalized reporting processes perceived as important motivation factors. This study contributes to organization and leadership studies and identifies problem areas, possibly helping managers and organizers focus further on the importance of anti-corruption work and whistleblowing processes within organizations. Further studies are recommended to increase the field of knowledge related to employees' willingness and motivation to notify when they witness workplace corruption.

**Keywords:** corruption; whistleblowing; organizational culture; reward

## 1. Introduction

Corruption exists at most levels of society, and occurs in both the public and private sector. Corruption can take many forms. There may be abuse of power, bribes, unfair favoritism, undue influence or camaraderie. Norway is, in an international context, viewed as a country with little, or close to no corruption [1]. There exists an assumption that organizations that display solid cultures and the right core values, also hold the ability to counteract unethical behavior within their organizations, and that they have in place well-elaborated notification instructions and routines, that encourages employees to report findings or suspicions of corruption [2]. The problem is, however, that a 2003–2016 review of Norwegian court-rulings indicated that corruption does occur [3] and that only 50% of employees that have witness or experienced corruption and/or wrongful conduct chose to report [4,5]. The consequences if employees fail to notify, is that corruption may develop and flourish [1]. Report failure may again threaten Norwegian democracy and welfare-society and hinder economic growth [6]. Failing to notify may further hurt individuals, as it might lead to wrongful administrative decisions or missing permissions [7]). Sources today are conflicting as to why employees fail to report. Reasons range from a lacking willingness to face professional reprisals [8],

a fear of experiencing workplace bullying [9], and a fright of falling victim to professional and personal isolation [10].

- *Aim and objectives*

Pondering the last years 'court-rulings there is, therefore, a need to examine the fear behind corruption-related whistleblowing at Norwegian workplaces. Especially, and what should be considered a gap in the literature, is there a need to investigate the relationship between notifications and reward [11]. The aim of this study was therefore to examine employee's perception of notification routines, and to add new knowledge to whether an offer of reward would affect their willingness to report. The following research questions were developed: (1) What are the employee's subjective perceptions of the municipality's notification routines? (2) What are the employees' subjective perceptions on type of reward, and (3) is there a correlation between offers of rewards and loyalty and employees' willingness to report? First, this paper starts out with a review of literature on corruption, whistleblowing, motivation and reward. Second, it outlines the study's materials and methods, before it thirdly summaries and presents the study's results. The discussion section then interprets and describes the significance of findings, before the last section summarizes the answers to the stated research questions, outlines implications and make recommendations.

## 1.1. Corruption

Corruption is a misuse of power for personal gain or a means of extortion to avoid disadvantages [12]. It is also present when the behavior of public officials deviates from accepted norms to achieve individual advantages [13]. International research on why employees chose to be corrupt has shown that their actions are either based on their own calculated choices or elements that may lay beyond their control [14]. Thus, possible reasons include psychological, political or ideological factors [15]. While Fijnaut and Huberts [16] argued that there were multiple reasons behind corruption in the public sector; individual, organizational, social, and economical, Rose-Ackerman [17] claimed that corruption was due to one thing only: an expectation of that the financial dividend will exceed the costs. Further, for corruption to occur, Jain [18] outlined three factors that had to be present. The first was the presence of power. There must be an authority that can both establish and enforce a new set of rules. Second, the power factor must be seen in the context of financial gain, and thirdly, the risk of being caught or to face extensive reactions must be considered low. Jain's findings were supported by Søreide [1], who stated that the probability of corruption increased if the decision-makers had a power monopoly, or parts of the decision-making process or outcome, could be kept secret.

## 1.2. Whistleblowing

A means to fight corruption is reporting, or whistleblowing, which Near & Miceli [19] defined as the reporting of wrongdoing that is either illegal, unethical or worthy of action in the workplace. There are conflicting reasons why employees choose to report on corruption. Cited reasons include the taking of social responsibility [20], the achievement of personal gain [21], and a desire for personal advantages [22]. Near and Miceli [23] concluded that very few whistleblowers were in fact malcontents seeking, or with an interest in spreading, false rumors. On the contrary, most of them were employees that had witnessed injustice and wished for it to stop. Despite this, the same research found that only the most serious and obvious wrongdoings in the workplace were reported. Generally, a whistleblowing process involves three main parties; the whistleblower, the accused, and the recipient [24]. Eriksen [1] pointed to the importance for managers in protecting the reporting employee from retaliation at the workplace. To create a sense of safety, such protective measures should be formalized, in writing, and be easily available to all [25]. The management should, therefore, strive to make the notification procedure clear, securely committed, and predictable. Thus, treating and considering it as an extraordinary intra-organizational reporting channel [26]. Upon receiving a report of concern, it is important, that the receiving end, often the management, process the right knowledge and attitudes. De Graaf [27] found that in organizations where the top-management emphasized professionalism and fairness, notification systems were perceived more

effective and workable than in organizations that signaled indifference or that noteworthy conditions were not highly ranked or prioritized.

### 1.3. Motivation and Reward Systems

A system of reward is here, designed to increase employees' willingness and motivation to notify when they witness workplace corruption. The goal is to put an end to wrongful conduct. Thus, motivation is a process that engages, directs, maintains and decides behavior intensity [28]. The scientific interest for motivation evolved largely during the 1970s as a response to behavioral psychology. The study of motivation seeks to reveal individual motivation factors and decision-making processes [29]. As what motivates differs from individual to individual, known literature on the subject often distinguishes between social theories, needs theories, job characteristics models and cognitive theories [28]. In social theories, the main motivational factor is a sense of perceived justice. Employees are generally concerned with fairness in the workplace. The partitioned of tasks and rewards will affect the worker's job motivation [30]. The staffs' perceived degree of task and rewards distribution is, however, not just uniform, it is also dependent on whom they compare themselves to [28]. If a worker feels unfairly treated, he or she will often try to create a balance by trying to influence the scope of the tasks or the size of the reward [31]. From a need-theory viewpoint, human needs can, as outlined in Maslow's [32] Hierarchy of Needs, be divided into five main categories; physiological, safety, love/belonging, esteem, and self-actualization. The basic idea is that the lower core needs have to be satisfied before the achievement of the higher ones. As further argued by Maslow and later supported by Alderfer's [33] Existence, Relatedness, and Growth theory, it takes a lot before we feel that our safety is threatened. In this context, a fear of retaliation from either co-workers or management can, however, get in the way of workplace motivation and development.

Job characteristics models focus on the execution of the tasks themselves, with a concentration on job features that influence worker motivation. Herzberg's [34] two-factor theory divides between factors that create worker satisfaction and factors that result in workplace dissatisfaction. Those who create satisfaction Herzberg referred to as motivation factors, examples being perceived sense of achievement and personal development. Examples of the opposites, or hygiene-factors, are perceived safety, physical and psychological working conditions. These dimensions are not negative in themselves but can lead to demotivation if they are lacking or perceived not present. In this context, an economic reward can be a hygiene-factor. In itself, not a prerequisite for a good working environment, but can serve as a motivation factor if present, and serve as a negative influence if taken away. While job characteristics models focus on work execution, the source of motivation from a cognitive viewpoint lies in expectations of goal achievement. In 2009, Kaufmann and Kaufmann [28] argued that employees will, in most cases, choose actions that bring about relevant results, and give the greatest rewards. These assumptions are based on a notion that workers perceive that their degree of effort equals reward, that workers will conduct a contemplated perceived reward calculation, and that they assume that their achieved rewards, will be in proportion to their actual effort made.

### 1.4. Reward and Whistleblowing

Reward can be a collective term for all forms of performance-specific remuneration, such as praise, perceived mastery, and benefits [31]. Intrinsic motivation often referrers to the behavior that is driven by a feeling of inner satisfaction, while extrinsic motivation is related to a desire for reward or aspiration to avoid punishment [29]. Armstrong [35] found that recognition was one of the most desired extrinsic motivation factors. That became especially evident in cases, which the employees felt that the management was responsive and implemented changes and gave recognition based on the employee's suggestions. While there are many positives to reward striving, it can also become negative. If the pursuit of reward gets too absorbing, it may displace the joy of task performance, which over time can affect the employee's intrinsic motivation [29]. Formalized systems and approaches to critical incident reporting vary internationally, and from organization to organization. The United States has developed a financial reward system for workers that report on critical incidents in the workplace. A prerequisite for reward is here that the ruling becomes final in the legal

system. The American legislation has thus two mechanisms in place in these cases; one to protect the individual whistleblower and one that regulates the level of reward [36]. In Norway, which is the case of interest here, there is not developed such a regulated reward system. However, there is no obstacle for organizations to develop their schemes if desired [1]. As Eriksen's [1] research indicates that propensity to notify increases with a formalized reward system, more research is needed on the motivation for notification in systems without a formalized reward system. Specially, is there a need to examine whether offers of reward, eventually what type of reward, that can influence employees' willingness and motivation to report if they witness or discover workplace corruption.

*1.5. Beyond Rewards—INDIVIDUAL Ethical Considerations*

The motivation or practice to report corruption or other perceived wrongdoings outside a workplace can also be interpreted in the terms personal ethical considerations or deliberation, which can be afflicted by a reward system. MacIntyre [37] argued that a practice is a "*coherent and complex form of socially established cooperative human activity through which goods internal to that form of activity are realized in the course of trying to achieve those standards of excellence which are appropriate to, and partially definitive of, that form of activity, with the result that human powers to achieve excellence, and human conceptions of the ends and goods involved, are systematically extended.*" (p. 187.). Within personal perspectives there are always choices, and in an organization, there are often multiple options. The individual must to some extent makes individual value judgments before reporting or using internal procedures or external resources. Such an ethical discourse can be seen along the two continuums of legal versus illegal and legitimate and illegitimate. We have expressed this theoretically in a descriptive four columns ethic model (Figure 1).

| Actions/practice | Legal | Illegal |
|---|---|---|
| Legitimate | (i) Not problematic, a no blow | (ii) Ethically problematic, |
| Illegitimate | (iii) Ethically problematic | (iv) Not problematic, blow! |

**Figure 1.** The Whistleblowers choice model**.**

- The whistleblower problematic choice model

In the organizational choice to be made can theoretically be divided into four options (i-iv). If a practice is perceived by the individual as both (i) legal and legitimate it arises no ethical concerns. The same goes for option (iv) given solid proof. The fields (ii-iii) is problematic because of the individual carry different notions of loyalty, foremost to their close working colleagues [38]. The loyalty to co-workers not receiving benefits but a loving relationship with a contractor (iii) and having conflicting interests. Loyalty towards the involved co-worker can delimit the chance of whistleblowing, but gradually along the axis of one-night stand to marriage. A breach in good practice perceived as illegal can also be that the ongoing practice accepts small tokens of goodwill such as a cake provides or a free lunch, but can increase strain when it comes to having your house rebuilt. Most organizational practices give some leeway along the continuums of legality and legitimization [39]. Before we present our findings, we will present how got to them.

**2. Materials and Methods**

This study reports on data collected from five focus-group interviews with together 14 employees (N = 14) belonging to a city planning department in a medium-sized municipally in the southeastern parts of Norway. This population sample was deemed relevant, as the Norwegian

Ministry of Justice, Public Security [6] and Transparency International Norway [2] had identified employees belonging to city planning departments as high-risk when it came to potentially being exposed to corruption inquiries, hence received 30 per cent of all queries in 2013 [2]. Data was collected during the spring of 2018. Besides one, all the groups consisted of three subjects. The selection of participant was based on an assumption that the employees belonging to a city planning was especially exposed to corruption. As documented by the Norwegian Ministry of Justice and Public Security [6] and Transparency International Norway [2], employees in city Norwegian planning departments have a history of and receives about 30% of all corruption inquiries. The application for permission to collect data was sent to the Norwegian Center for Research Data (NSD). Request to research the organization was sent to the municipality in question. Both gave their consent. The participants received an invitation letter by email. The invitation outlined the research project and emphasized volunteerism, the ability to withdraw, and ensured anonymity at all stages of the process.

*2.1. Data Collection and Procedures*

The collection of data occurred through focus-group interviews. The location was on municipally ground, in a separate room reserved for the occasion. An interview-guide had been developed (Table 1) based on materials from existing research. The first section of the interview (questions 4-6), which was related to the participant's knowledge and perception of the municipality's notification routines, and based on a previously developed interview guide by Matthiesen et al. [4] and Trygstad [40]. The focus here was not to test or gain detailed information about the structure and the content of the municipally's notification system and routines, but rather reveal whether the participants knew of it and if so, their subjective perceptions. On that note, it should be mentioned that the municipally in this study is a Transparency International (TI) member, has developed a notification system based on an international TI standard. Further, all line-managers have been trained and completed course-work in anti-corruption work. The second part of the interview (after question 7), presented the participants with a relevant case and questions inspired by Matthiesen et al. [4] study on wrongful conduct and notification in Norwegian working life, and Mansbach and Bachner's [41] study "Internal or external whistleblowing: Nurses' willingness to report wrongdoing". The study's findings indicated, among others, that only half of those who had witnessed corruption reported their observations, that there existed a fear of reporting and that whistleblowing largely had been ignored in public sector. The last section asked whether an offer of compensation would have an impact on employees' willingness to notify corruption. These questions were based on questions earlier developed by Solum and Andås [11]. Before the main interviews, a pilot-interview was conducted to ensure clarity and eliminate problems and barriers [42]. A revised and final version was then developed based on feedback. All interviews were audiotaped and conducted in one session. The researcher served as both interviewer and moderator. In the beginning, the participants were introduced to the researcher and received a short structure briefing. Volunteerism and privacy were again emphasized. All questions were asked in order, and participants were given the possibility to take notes. All inputs and subjective opinions were throughout the interview discussed openly and together. In the end, the participants were allowed to clarify previous statements and make final remarks about the interview process. The interviews lasted for about forty minutes each.

- Case presented (after question 7):

*"You are at a work-related Christmas party. You overhear a conversation between some consultants representing two major construction companies - "COMPANY A" and "COMPANY B". The conversation revolves around how to acquire necessary building permits. The consultants believe these can be obtained by offering the assigned municipal caseworker "Ole", which they know just bought a new house, some free construction services. Further, it becomes evident from the conversation that COMPANY B is already in the process of developing a housing project in an area that is not regulated for residential purposes."*

**Table 1.** Interview guide.

| Q | | Questions |
|---|---|---|
| **1** | D | Gender |
| 2 | D | Years of professional experience |
| 3 | D | Age |
| 4 | K | Are you familiar with the notification procedures? |
| 5 | K | If yes, where did you learn about the notification procedures? |
| 6 | K | How do you perceive the opportunity to report wrongdoing? |
| 7 | P | Would you report the conversation you heard at the Christmas party? |
| 8 | P | If yes, would you report through internal or external channels? |
| 9 | P | What is your perception of "reward"? |
| 10 | P | What type of reward is important to you if you choose to report? |
| 11 | W | If rewards were handed out upon reporting, what type of rewards should be offered? |
| 12 | W | If reporting were rewarded in some way, would this affect your willingness? |
| 12 | W | If notification-rewards were formally introduced at your workplace, would you accept? |
| 13 | W | If the word "Reward" were replaced by the word "Compensation," would that affect your willingness to report wrongdoings? |
| 14 | W | How do perceive reporting will affect collaboration and the overall work-environment? |

Note: D = demography, K = knowledge of routines, P = perception of reward, W = willingness.

## *2.2. Analysis*

The analysis process started by transcribing each interview separately. To ensure a systematic approach, the researcher conducted a mapping of the individual response structures and divided them into sub-categories based on the interview-guide outline [43]. The main categories were (1) Distribution and demographics, (2) the participant's knowledge and perception of the municipality's notification routines, (3) the employees' subjective perception of the type of reward that was most desired, and (4) Whether an offer of compensation would have an impact on employees' increased willingness to notify corruption. The participant's responses were used as a basis of documentation.

## *2.3. Ethics*

Several steps were taken to ensure that all phases of this study were conducted in accordance with the ethical standards required and expected by the Norwegian Center for Research Data (NSD). NSD approval was sought and given before any data collection (ref. 57413). Participation was voluntary and no financial or other compensation was provided. Permission was obtained from the municipality top management. Participants were before the interviews provided with an information letter. Voluntarism, opportunity to withdraw and anonymity was assured throughout the process.

## *2.4. Limitations*

This study has limitations. One was that it was limited in scope as data was only collected from employees in only one municipality. Second, it did not, as a contributing reporting factor, consider the sample's perceived fear of work-place reprisals. Third, it did not pre-define associated terms like ethics and moral prior to the interviews, which left room for individual interpretation and, in turn, may cause somewhat lower term validity.

## **3. Results**

This section outlines a structured description of findings. This study was limited in scope as data was only collected from employees in only one municipality. Further, it did not, as a contributing reporting factor, consider the sample's perceived fear of work-place reprisals.

### 3.1. Distribution and Demographics

Fifteen employees agreed to participate in the study. Fourteen showed up for the interviews. Nine men and five women. Their ages ranged from 36 to 67. Seven of them had worked for less than five years, while four had ten or more years of professional experience.

### 3.2. Participant's Knowledge and Perception of the Municipality's Notification Routines

Summed, eight participants did not know the municipality's notification routines, while four responded affirmatively. The remaining two were unsure. Several participants expressed the feeling that reporting wrongdoings were not a prioritized area at the workplace. One participant stated "I have worked in public administration for many years, but have not learned how to-, or the importance of reporting wrongdoings. That is food for thought". Several found it likely that there existed a notification routine, but they had never read it or had no knowledge of where to find it. One stated, "There are no formalized routines here for those such things…like they have at other workplaces…maybe we have one somewhere… I don't know". When it came to channels for notification and whom to go to, the participants could point out several. Examples being their immediate supervisors and labor union representatives.

However, as one pointed out, "If the person doing wrong happens to be my immediate supervisor and labor union representative, I have no idea whom to turn to". It will be all up to the individual's ethical considerations. Several of the participants agreed that too much responsibility was left on the notifier and that it did not feel safe. As stated, "It feels like it is up to the notifier to provide solid evidence of wrongdoings and possible corruption". Another agreed and said, "You can lose your job, be excluded, or be sat to perform tasks that have so little utility value that they, in reality, equal being fired". Most participants agreed that reporting was the moral right to do, but here it was a gap between theory and practice. For example, several participants did feel that reporting suspicion of wrongdoing would be inhibitory for their further careers. A relatively new employee stated, "As I perceive the municipality, I would not dare to report my suspicions… As new, I would just hope that someone else took care of it… I would just withdraw from the entire situation". When it came to the participant's perception of how a possible reward upon reporting would impact their working environment, most believed it would have a negative impact. One said, "It might turn into a mass-surveillance culture…that is negative, as people may start to report wrongfully". However, as a couple pointed out, they felt it was a difference between `selling-out 'and reporting suspicion of wrongdoing. One said "It is not `selling-out` if you witness an act of corruption…then you should report…that's the right thing to do".

### 3.3. Participant's Perception of What Kind of Rewards They Considered Most Desirable

Multiple types of desired rewards were highlighted. Many felt that money or higher salary was most desirable. One said "In my case, money would be the desirable mean of compensation... I need money to live a decent life…that is just how it is right or wrong". Another somewhat agreed and stated, "To be honest, I would prefer money…just as everybody else". Others however disagreed and felt that making a profit on reporting was wrongful. As said, "To make money on reporting wrongdoing is almost as bad as performing the act itself…it just doesn't feel right". While money was viewed as the main motivator, several participants also pointed to the importance of being heard, valued and respected. One said "That you get recognized. That you receive positive feedback and honorable mention rather than being talked down is very important". Another argued, "The management needs to communicate that reporting wrongdoing is valued, that it is appreciated". Besides individual gain, contributing to good work ethics through reporting was also lifted as the desired reward. As one said "Being a `sell-out 'is an outdated term, that continues to live on. No one

likes a snitch. Therefore, when someone dares to report, there should always be someone ready to chair him or her on… if you take on a dirty job and nobody notices…. You will never do it again". Upon asked whether replacing the word "reward" with "compensation" would affect their willingness to report, the participants did not feel that that would make a significant difference. However, a couple pointed to how the word "compensation" more strongly reflected how reporting was considered a burden, and that experiencing a burden was something that should be compensated. One argued, "It means that it is a burden to report. That it is not going to be very comfortable…that it is something that requires compensation. I believe renaming it will work against its purpose".

*3.4. Participant's Perceived Relationship between Compensation and the Willingness to Notify*

Other findings indicated that there were several factors associated with the reward that influenced the participant's willingness to notify. Regardless of the type of reward, several pointed to how recognition was the main motivator. As one pointed out, "If we had a system in place that recognized reporting of wrongdoing, we would have come a long way". Another said, "If you know you're being heard, you might start reporting other types of concerns also". Multiple participants agreed that introducing reward would have a strong symbolic effect. As one argued "Regardless of the reward being money or not…a reward says something about how the employer views and appreciates reports of wrongdoing. That reflects the workplace in all areas but one. Perceived recognition is important". On that note, money was still lifted as an important motivator, especially if the sum was viewed considerably. As argued, "We are just simple human beings. Just regardless, we get influenced by money offerings, consciously or subconsciously".

Also, there was expressed a fear that if the amount offered was too high, it may lead to an over-willingness to report. One said, "It might increase the probability of wrongful reporting. If the desire for reward becomes a motivational factor, it might lead to a search for non-existing correlations". During group discussions, the participants also divided between clear violations and "grey-zone" cases. Most agreed that e.g., the case they had been presented was not just black or white. While several would report clear infringements, they were more unsure what to do when encountering "grey-zones". Here, some believed that an offer of compensation would have a positive effect. In total, 12 out of 14 participants would have accepted an offer of compensation if it were enshrined in the municipality's routines. An important prerequisite was that the process was then formalized and equal to all. Only two participants were not willing to accept any form of compensation upon reporting. As argued, "I find it extremely troublesome that people are bribed to report" and "I find it problematic that one should accept a reward to report wrongdoing in the first place". However, they both agreed that if the reward were a type of formalized, ensured care and respect they would accept it.

**4. Discussion**

The aim of this research was to examine corruption-related whistleblowing at Norwegian workplaces. Especially, a goal was to contribute to closing the literature gap related to the relationship between notifications and reward. This section systematically interprets and describes the significance of findings, and discuss how results can be interpreted in perspectives of earlier outlined theoretical frameworks and within the stated scope of this study. The analysis indicated lacking knowledge and perception of the municipality's notification routines. This affected the employees' views- and willingness to report. In this study, several subjects assumed there existed a formalized routine, but they were not familiar with its contents when to use it, nor where to find it. This leaves it to the individual ethical consideration and supports that whistleblowing may be related to altruistic ideals. It possibly also indicates a lack of focus in management distribution, and thus supports McCoy and Heckel's [44] earlier findings which indicated that having a managerial focus on the importance of reporting wrongdoings at the workplace, positively contributed to the worker's knowledge and perception. A contrasting finding here, however, is how the interview subjects in this study, while displaying lacking notification knowledge, simultaneously identified several perceived

arenas for notification, e.g., their immediate supervisor, the Human Resource Director and their Labor Union representative. Traditionally that symbolizes an open, transparent and including workplace environment [45].

Another important discovery was related to the participant's notion of security. The subjects found it difficult to file a report on a co-worker, as it was perceived to be uncomfortable, and associated with a great deal of uncertainty related to whether the provided information was correct or not. This coincides with previous study findings which showed that whistleblowing is often viewed as personally distressing [9], associated with a fear of workplace retaliation [22], and a perceived lack of security feeling [1]. Fear of lacking security became especially evident when they discussed whether they would report an immediate supervisor or another municipal superior. Here, several of the participants exhibited a fear of workplace retaliation, which are consistent with earlier research findings [5,22,46]. This can be seen as both obedience and loyalty.

When it came to the participant's perception of what kind of rewards they considered most desirable, this study found that money was not the main motivating factor. Instead, a perceived sense of self, workplace recognition and a notion of being heard were considered more desirable and possibly related to altruism and high vocational ethical standards. The notion of accepting a monetary reward was considered immoral by several people, a continuation of the wrongdoing itself, and somewhat contradictory to being a responsible public servant [47]. Several participants expressed, as earlier indicated by Armstrong [35], how increased recognition and respect from the top-management in cases of whistleblowing could serve as an important motivation factor, hence create more willingness to report. Further, the participants that expressed somewhat resistance towards money rewards viewed symbolic rewards, e.g., extra vacation days with pay as more desirable, as that would be, in their views, more righteous, morally correct and with greater intrinsic value.

A sense of righteousness also influenced the participant's perceived relationship between compensation and their willingness to notify. In this study, participants who expressed high morals perceived themselves as more willing to report wrongdoings, as it, according to them, was "the right thing to do". Thus, they viewed whistleblowing as an altruistic act regardless of reward or compensation [20]. Others, however, based on a pragmatic cost-benefit thinking, projected that their willingness to notify would increase if they were offered some type of compensation for the possible displeasure and social stress they could encounter as a result of whistleblowing. On that note, the participants also warned about creating a too formalized and too comprehensive reward system, as it may lead to incorrect reporting, where the focus becomes more on the prize itself, rather than on reporting and preventing unwanted behavior [48]. However, overall, findings indicated that almost all participants would accept an offer of reward or compensation if it was formalized and included in the municipality's frameworks and routines. The main argument here, in support of Adams, [30], was a feeling of workplace acceptance and justice.

## 5. Conclusions, Implications, and Recommendations

### *5.1. Conclusion*

Overall, the participants had limited knowledge of the municipality's formalized notification routines. However, most knew of several informal arenas for notification of wrongdoings, e.g., their immediate supervisor, the Human Resource Director and their Labor Union representative. Despite a perceived good working environment, most found it difficult to report, as they feared workplace retaliation, personal distress and a lack of security. This can be related to legal/legitimization ambiguity. Further findings indicate that the employees felt that the municipality top-management did not facilitate nor encouraged whistleblowing. Nor did the management have a sufficient perceived focus on ethical guideline development and anti-corruption work. When it came to rewarding, most subjects were opposed to receiving money. Instead, symbolic rewards such as extra days off with pay increased perceived sense of self, workplace recognition and a notion of being heard were considered more desirable. Overall, findings indicated that multiple factors influenced

the employee's willingness to report and receive compensation. In particular, increased management recognition and a more clearly formalized reporting processes were perceived as important motivation factors.

*5.2. Theoretical and Practical Implications*

This study has both theoretical and practical implications. Theoretically, it contributes to the fields of organization and leadership studies. From a practical viewpoint, it also identifies problem areas, possibly helping public bodies, hence managers and organizers, focus further on the importance of anti-corruption work and whistleblowing processes within organizations.

*5.3. Recommendations*

This study recommends an increased focus and development of corruption notification routines in Norwegian municipalities. Further studies are recommended to increase the field of knowledge related to employees' willingness and motivation to notify authorities when they witness workplace corruption.

**Author Contributions:** Conceptualization, J.L.S. and A.M.N.G.; methodology, J.L.S. and A.M.N.G. formal analysis, J.L.S. and A.M.N.G.; investigation, A.M.N.G.; data curation, A.M.N.G..; writing—original draft preparation, A.M.N.G.; writing—review and editing, J.L.S. and L.I.M.; visualization, J.L.S. and L.I.M.; supervision, J.L.S.; project administration, A.M.N.G. and J.L.S. All authors have read and agreed to the published version of the manuscript.

**Funding:** This research received no external funding

**Acknowledgments:** Reported findings were first published in: Gaup, A.M.N. (2018). How can whistleblowing be rewarded? [Hvordan kan varsling belønnes?] (Master-thesis). University of South-Eastern Norway. USN School of Business, Norway.

**Conflicts of Interest:** The authors declare no conflict of interest.

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
