# Peer review of "Whistleblowing in Norwegian Municipalities—Can Offers of Reward Influence Employees’ Willingness and Motivation to Report Wrongdoings?"

_sustainability, doi:10.3390/su12083479_

Round 1
Reviewer 1 Report
Suggested Revisions
It would be useful to start the introduction by describing the broad context of the study.
After describing the different existing approaches on the topic, the authors should outline the gap (those aspects which are no so well studied and could be of interest) and introduce the aim of the paper.
Typically, at the end of the introduction, there is presented the structure of the rest of the paper.
In the conclusion section, should be outlined the implications of the study
More information should be provided on the participants' selection.
Line 209-215: Please provide some explanations on the Case presented (after question 7)
The text of the paper should be structured into paragraphs.
Please review the reference style according to with MDPI reference list and citations style guide https://www.mdpi.com/authors/references.
See also the MDPI guidance for text formatting and how the figures/tables should be numbered.
Author Response
Dear Reviewer 1,
Thank you for taking the time to review this paper and for providing much useful and valued feedback. We have made the following changes in the document based on your feedback:
It would be useful to start the introduction by describing the broad context of the study.
The start of the introduction now reads (line 30): “Corruption exists at most levels of society, and occurs in both public and private sector. Corruption can take many forms. There may be abuse of power, bribes, unfair favoritism, undue influence or camaraderie. Norway is, in an international context, viewed as a country with little, or close to no corruption [1]”. The added text is from the same reference [1].
After describing the different existing approaches on the topic, the authors should outline the gap (those aspects which are no so well studied and could be of interest) and introduce the aim of the paper.
The new paragraph reads (line 45): “Pondering the last years ‘court-rulings, there is therefore, a need to examine the fear behind corruption-related whistleblowing at Norwegian workplaces. Especially, and what should be considered a gap in the literature, is there a need to investigate the relationship between notifications and reward [11]. The aim of this study was therefore to examine employee’s perception of notification routines, and to add new knowledge to whether an offer of reward would affect their willingness to report. The following research questions were developed: 1) What are the employee`s subjective perceptions of the municipality`s notification routines? 2) What are the employees' subjective perceptions on type of reward, and 3) is there a correlation between offers of rewards and loyalty and employees' willingness to report?”
Typically, at the end of the introduction, there is presented the structure of the rest of the paper.
The last section of the introduction now reads (line 53): “First, this paper starts out with a review of literature on corruption, whistleblowing, motivation and reward. Second, it outlines the study`s materials and methods, before it thirdly summaries and presents the study`s results. The discussion section then interprets and describes the significance of findings, before the last section summarizes the answers to the stated research questions, outlines implications and make recommendations.”
In the conclusion section, should be outlined the implications of the study
The section now reads (line 387): This study has both theoretical and practical implications. Theoretically, it contributes to the fields of organization and leadership studies. From a practical viewpoint, it also identifies problem areas, possibly helping managers and organizers focus further on the importance of anti-corruption work and whistleblowing processes within organizations”
More information should be provided on the participants' selection.
The new sections now reads (line 185): “This study reports on data collected from five focus-group interviews with together 14 employees (N=14) belonging to a city planning department in a medium-sized municipally in the southeastern parts of Norway. This population sample was deemed relevant, as the Norwegian Ministry of Justice, Public Security [6] and Transparency International Norway [2] had identified employees belonging to city planning departments as high-risk when it came to potentially being exposed to corruption inquiries, hence received 30 per cent of all queries in 2013 [2].”
Line 209-215: Please provide some explanations on the Case presented (after question 7)
The new section now reads (line 207): “The second part of the interview (after question 7), presented the participants with a relevant case and questions inspired by Matthiesen et al. [4] study on wrongful conduct and notification in Norwegian working life, and Mansbach and Bachner's [41] study “Internal or external whistleblowing: Nurses’ willingness to report wrongdoing”. The studies` findings indicated, among others, that only half of those who had witnessed corruption reported their observations, that there existed a fear of reporting and that whistleblowing largely had been ignored in public sector.”
The text of the paper should be structured into paragraphs.
Thank you. Please see changes
Please review the reference style according to with MDPI reference list and citations style guide https://www.mdpi.com/authors/references. See also the MDPI guidance for text formatting and how the figures/tables should be numbered.
Thank you. Please see changes
Reviewer 2 Report
I congratulate the authors for carrying out the research behind the article. It is a very important topic to deal with if we want to build up a healthy society.
Some advices on the structure of the article:
- In the introduction, there is a reference about the “personal” problems that the whistle-blowers could face if they report, but there is no research question on this (although in the findings we can find collected opinions on this). From this description, I think that the research questions must take this into account.
- The research objectives demand a separate section; I do not think that it is advisable to be included in the introduction.
- I think that the case description and the subsequent table could be sent to an annex.
- As stated in line 361, Section 4 instead of placing a Discussion is much more a structured description of the Findings (my suggestion for the title).
- So I miss a real section for Discussion.
- In my opinion, Section 5 should be split into three sections: Conclusions, Implications for Public Bodies, and Implications for Academic Research.
- In line 353 there is a reference to “participants with high expressed moral” and in the paper there is no definition of this concept or a scale to measure it.
- I think that it is needed some kind of description of the notification routines of the municipality and the rewards it offers, because some of the lack of knowledge of the participants in the focus-groups may be related to that (there can be a Section on this).
Some comments on grammar and style:
- Even if English is not my mother tongue is not English, I think that all the sentences need a Subject; and the Verb follows the Subject without comma (some commas precede verbs without being needed). I will point out some cases, but I strongly advise the authors to bear this in mind while revising the paper.
- Abstract needs grammatical revision: Line 17, municipally?; Line 19 “Especially was increased…”.
- Line 36: witnessed.
- Line 37, “comma before is” is not needed.
- Line 65, citation needs to be checked.
- Line 97: partitioned?
- Line 121: assumptions ARE based.
- Line 137: citation is needed for the US financial reward system, if it Is [36] I think it is better to indicate in this sentence (and not in the next one).
- Line 145: There is a need.
- Line 150: in terms of personal… (“the” is not needed and I miss “of”)
- Line 162: with the figure, it is usual to citate the source (even if the authors are the source)
- Line 164 to 174, description of the Figure A1: I advise a revision of the style. It is difficult to understand.
- Line 183: revise redaction (verbs).
- Line 193: “, which” are not needed.
- Line 195: there is a bracket, but not the final one. Revise redaction.
- Line 204: structured.
- In Line 185 and Line 186 NSD is mentioned in the same way as in Line 231; I advise to keep this last one only.
- Line 250: after several, instead of a final point, I recommend a semicolon.
- Line 259: several participants did
- Line 319: lack of
- Line 332: o
- Line 377 and 378: “Especially was increased”. Revise redaction.
- The authors use Especially and Specially, both correct, but I do not see the point of using in the same article both words.
Author Response
Dear Reviewer 2,
Thank you for taking the time reviewing our manuscript and for providing valuable feedback. We have made the following adjustments based on your comments:
In the introduction, there is a reference about the “personal” problems that the whistle-blowers could face if they report, but there is no research question on this (although in the findings we can find collected opinions on this). From this description, I think that the research questions must take this into account.
Thank you for this insight. We do agree with your line of argument. As the aim of this study was to examine employee’s perception of notification routines, and to add new knowledge to whether an offer of reward would affect their willingness to report we did not include/focus on the sample`s perception on possible reprisals. We have therefore, based on your feedback, included this as a limitation (see new section 2.4.)
The research objectives demand a separate section; I do not think that it is advisable to be included in the introduction.
Thank you. Aim and objectives are now organized under a separate heading (line 47).
I think that the case description and the subsequent table could be sent to an annex.
Thank you. We included the above mentioned in the main text as the manuscript preparation guidelines states that “All Figures, Schemes and Tables should be inserted into the main text close to their first citation and must be numbered following their number of appearance (Figure 1, Scheme I, Figure 2, Scheme II, Table 1, etc.).”. However, we do agree that both indeed could be sent to an annex, so we are more than happy to chance this if editor approves.
As stated in line 361, Section 4 instead of placing a Discussion is much more a structured description of the Findings (my suggestion for the title). So I miss a real section for Discussion.
Thank you for addressing this issue. We have added a section to the beginning of Section 4 explaining the outline and purpose of this section. Further, we have integrated a line about a structured description of findings in Section 3 where the results/findings are presented. We have discussed back and forth, but find it a little hard to grasp what you mean by “So I miss a real section for discussion”. Hopefully, the added section contributes to clarity.
In my opinion, Section 5 should be split into three sections: Conclusions, Implications for Public Bodies, and Implications for Academic Research.
Thank you. We have now spit Section 5 into sub-sections. As multiple reviewers provided valued feedback on the formatting of this section, we have tried to incorporate all your inputs. This sections is now divided into 1) conclusion, 2) Theoretical and practical implications, and 3) recommendations.
In line 353 there is a reference to “participants with high expressed moral” and in the paper there is no definition of this concept or a scale to measure it.
Thank you for this observation. We agree. We did not pre-define terms like moral and ethics prior to the interviews. We see that this may cause somewhat lower term validity. We have added a section 2.4. Limitations, where we address this issue together with other relevant limitations.
I think that it is needed some kind of description of the notification routines of the municipality and the rewards it offers, because some of the lack of knowledge of the participants in the focus-groups may be related to that (there can be a Section on this).
We agree. We have added the following section in section 2.1.: “The first section of the interview (questions 4-6), which was related to the participant`s knowledge and perception of the municipality's notification routines, and based on a previously developed interview guide by Matthiesen et al. [4] and Trygstad [40]. The focus here was not to test or gain detailed information about the structure and the content of the municipally` s notification system and routines, but rather reveal whether the participants knew of it and if so, their subjective perceptions. On that note, it should be mentioned that the municipally in this study is a Transparency International (TI) member, has developed a notification system based on an international TI standard. Further, all line-managers have been trained and completed course-work in anti-corruption work.”
Some comments on grammar and style: Even if English is not my mother tongue is not English, I think that all the sentences need a Subject; and the Verb follows the Subject without comma (some commas precede verbs without being needed). I will point out some cases, but I strongly advise the authors to bear this in mind while revising the paper.
Thank you. All feedback that contributes to enhance our paper is highly appreciated. We have submitted the manuscript to a professional language editor.
Reviewer 3 Report
Dear Authors,
Thank you very much for giving me the chance to read your paper about whistleblowing and rewards in the public context. The topic is relevant to the actual debate. However, it has already received attention in previous research, especially a vast literature about rewards and whistleblowing already exists, and it is not considered in your literature review.
I think that while the topic of this study is of potential interest, its overall contribution to the field appears too limited for the broad readership of the journal for three main reasons.
First, the paper has no traces of relation with gaps related to sustainability in none of its conceptualizations. Therefore, I think that has little sense publishing it in a journal devoted explicitly to sustainability-related topics and research. In my opinion, there is no fitting between your paper and the journal's aims and scope.
Second, it is tough to evaluate the level of contribution and novelty of your paper to the overall literature given an in-depth and structure literature review about rewards and whistleblowing is missing in your research. A lot of literature about rewards and whistleblowing is already published and not considered in your research; therefore, it is not clear if the evidence emerging from your research can be considered a novel contribution or not. See some examples of missing references attached to the bottom.
Thirds, the paper is mainly empirical, and its contributions to the theory are questionable, given the research questions are remarkably related to the subjective perceptions of municipalities employees. However, the findings seem to add little to the overall theoretical framework of whistleblowing and rewards. Probably more structured multiple case studies with a more structured analysis of empirical results can provide some novel contributions, once the literature review about whistleblowing and rewards is provided, without it any empirical evidence risk to be overlapping with the previous studies.
Given the weaknesses of the study are too relevant to configure a major revision, I reject the study in the present form. However, I think that you could resubmit once you have strengthened the connections with sustainability topics, developed an in-depth literature review, and improved the research design in terms of multiple sources of empirical evidence and a more structured analysis of data.
References
Ayagre, P., & Aidoo-Buameh, J. (2014). Whistleblower reward system implementation effects on whistleblowing in organisations. European Journal of Accouting, Auditing, and Finance Research, 2(1), 80-90.
Bouville, M. (2008). Whistle-blowing and morality. Journal of business ethics, 81(3), 579-585.
Brink, A. G., Lowe, D. J., & Victoravich, L. M. (2013). The effect of evidence strength and internal rewards on intentions to report fraud in the Dodd-Frank regulatory environment. Auditing: A Journal of Practice & Theory, 32(3), 87-104.
Carson, T. L., Verdu, M. E., & Wokutch, R. E. (2008). Whistleblowing for profit: An ethical analysis of the Federal False Claims Act. Journal of Business Ethics, 77(3), 361.
Clark, K., & Moore, N. J. (2015). Financial Rewards for Whistleblowing Lawyers. BCL Rev., 56, 1697.
Dasgupta, S., & Kesharwani, A. (2010). Whistleblowing: a survey of literature. The IUP Journal of Corporate Governance, 9(4), 57-70.
Gao, L., & Brink, A. G. (2017). Whistleblowing studies in accounting research: A review of experimental studies on the determinants of whistleblowing. Journal of Accounting Literature, 38, 1-13.
Givati, Y. (2016). A theory of whistleblower rewards. The Journal of Legal Studies, 45(1), 43-72.
Iwasaki, M. (2018). Effects of External Whistleblower Rewards on Internal Reporting. Harvard John M. Olin Fellow’s Discussion Paper Series, (76).
Latan, H., Jabbour, C. J. C., & de Sousa Jabbour, A. B. L. (2019). ‘Whistleblowing Triangle’: Framework and Empirical Evidence. Journal of Business Ethics, 160(1), 189-204.
Lee, G., & Turner, M. J. (2017). Do government administered financial rewards undermine firms' internal whistleblowing systems?. Accounting Research Journal.
Miceli, M. P., & Near, J. P. (1994). Whistleblowing: Reaping the benefits. Academy of Management Perspectives, 8(3), 65-72.
Rose, J. M., Brink, A. G., & Norman, C. S. (2018). The effects of compensation structures and monetary rewards on managers’ decisions to blow the whistle. Journal of Business Ethics, 150(3), 853-862.
Author Response
Dear Reviewer 3,
Thank you for taking the time to read our manuscript and providing valuable feedback. We have based on feedback from You and the other reviewers made changes to the document. A main goal has been to clarify the aims and scope of the study, hence identify existing literature gaps and provide evidence of relevance. It is our belief that the paper now has a good connection with sustainability aims.
With regards,
Round 2
Reviewer 1 Report
I appreciate the efforts of the authors to improve the paper.
Reviewer 3 Report
Dear Authors,
Even if I appreciate your efforts to address my concerns, I think that the parts (I imagine the red lines) you have revised are not enough to address my doubts.
First of all, none of the revisions has addressed my concern regarding which is the relation between your paper and gaps related to sustainability in none of its conceptualizations. Also, in this revised version of the paper, there is no fitting between the paper and the journal's aims and scope given you never mention sustainability or the link between your paper and any of the different conceptualizations of sustainability.
Second, also the issue of evaluating the level of contribution and novelty of your paper is not addressed given in my opinion; there is the need for an in-depth and structured literature review with review tables that highlight the previous findings concerning whistleblowing and rewards. Moreover, none of the suggested references are taken into consideration to improve the reporting of previous findings.
Thirds, the paper is still mainly empirical, and its contributions to the theory are questionable, given the chosen empirical setting is remarkably related to the subjective perceptions of Norwegian municipalities’ employees, and the findings add little to the overall theoretical framework of whistleblowing and rewards. I am sorry to restate my previous doubts, but your revised version does not address properly any of them.